# Identifying Brain Abnormalities with Schizophrenia Based on a Hybrid Feature Selection Technology

**Chen Qiao [1],\*, Lujia Lu [1], Lan Yang [1] and Paul J. Kennedy [2]** 

[1]  School of Mathematics and Statistics, Xi'an Jiaotong University, Xi'an 710049, China; luke2016@stu.xjtu.edu.cn (L.L.); yanglan2018@stu.xjtu.edu.cn (L.Y.)

[2]  Center for Artificial Intelligence, University of Technology Sydney, Sydney 2007, Australia; Paul.Kennedy@uts.edu.au

\*  Correspondence: qiaochen@xjtu.edu.cn; Tel.: +86-029-82660949

**Featured Application: The hybrid feature selection method, which combines both machine learning and traditional statistical methods, is proposed to identify the brain abnormalities of schizophrenia. The results suggest that the brain regions and connectivity in SZs are destroyed compared with HCs, which may cause the cognitive deficits and autistic thinking in SZs. The findings support the validation of the proposed hybrid feature selection method, and thus, it is promised that such a hybrid feature selection method can be further used for other kinds of medical data analysis to enhance the diagnosis ability and further for precision medicine.**

**Abstract:** Many medical imaging data, especially the magnetic resonance imaging (MRI) data, usually have a small sample size, but a large number of features. How to reduce effectively the data dimension and locate accurately the biomarkers from such kinds of data are quite crucial for diagnosis and further precision medicine. In this paper, we propose a hybrid feature selection method based on machine learning and traditional statistical approaches and explore the brain abnormalities of schizophrenia by using the functional and structural MRI data. The results show that the abnormal brain regions are mainly distributed in the supramarginal gyrus, cingulate gyrus, frontal gyrus, precuneus and caudate, and the abnormal functional connections are related to the caudate nucleus, insula and rolandic operculum. In addition, some complex network analyses based on graph theory are utilized on the functional connection data, and the results demonstrate that the located abnormal functional connections in brain can distinguish schizophrenia patients from healthy controls. The identified abnormalities in brain with schizophrenia by the proposed hybrid feature selection method show that there do exist some abnormal brain regions and abnormal disruption of the network segregation and network integration for schizophrenia, and these changes may lead to inaccurate and inefficient information processing and synthesis in the brain, which provide further evidence for the cognitive dysmetria of schizophrenia.

**Keywords:** magnetic resonance imaging; schizophrenia; feature selection; brain abnormalities; biomarkers

## 1. Introduction

Schizophrenia (SZ) is a kind of mental disorder characterized by abnormal social behaviour and a failure to understand reality. Recently, decades of research on brain structure and function have provided us with some understanding of the neurobiological mechanisms underlying its symptoms [1,2]. For example, studies on brain structure suggest that neuroanatomical alterations may underlie the clinical onset of psychotic symptoms. The findings from functional brain imaging studies support a leading hypothesis that SZ stems from disconnectivity, namely abnormal interactions

between wide-spread brain networks. Recently, neuroimaging techniques like structural magnetic resonance imaging (sMRI) and functional magnetic resonance imaging (fMRI) have become a powerful tool to examine the abnormal regions and aberrant connectivity of brain networks in SZ, which bring psychiatry from subjective descriptive classification into objective and tangible brain-based measures [3]. For example, Du et al. applied a novel group information guided method to estimate inherent dynamic functional brain networks and found that the abnormalities of SZ were mainly distributed in the cerebellum, frontal cortex, thalamus and temporal cortex [4]. With fMRI data of SZ, Shine et al. showed that dynamic changes of functional connectivity are essential for cognitive processing [5]. Rosenberg et al. demonstrated that the whole-brain functional connectivity strength might serve as a biomarker of sustained attention for both healthy and disease assessments [6]. It was shown that functional connectivity profiles can predict levels of fluid intelligence [7]. By a supervised learning strategy that fuses sMRI, as well as fMRI data, some modality-specific biomarkers of generalized cognition with SZ were identified [1]. Based on sMRI data, Palaniyappan et al. suggested that concomitant increase and decrease in grey matter occur in association with persistent negative thought disorder in clinically stable individuals with SZ [8]. These studies on developing biomarkers allow the field of imaging analysis and psychiatry to move forward.

Given that SZ is often accompanied by cognitive decline, the thorough investigation of brain dynamics, as well as brain structure in SZ seems important in order to better understand the underlying neural mechanism. However, for MRI data of SZ, they usually have a small sample size, but a large number of features, i.e., $n \ll p$, where $n$ is the sample size and $p$ is the number of features [9]. For such kinds of data, there still lacks a systematic methodology to study them. That is because it is too difficult to discover the potential information contained in the data from a limited number of observations, which form a cognitive concept of the data or complete identification task [10]. To deal with data of dimensions much larger than the sample size, the generally used approach is dimensionality reduction. Feature selection and feature extraction are common methods for dimensionality reduction. For feature selection, those essential features of the raw data that have the greatest contribution to distinguish different objects can be identified. Thus, by feature selection, we can enhance the interpretability of learning, which is crucial for exploring the mechanisms of why things are different. Mathematically, consider any raw data as an $N$-dimensional vector $X = (x_1, x_2, \cdots, x_N)^T$, from which we can select $M$ features $\tilde{X} = (x_{R_1}, x_{R_2}, \cdots, x_{R_M})^T$ as required, where $x_{R_i}, i = 1, 2, \cdots, M$ are features chosen from $\{x_1, x_2, \cdots, x_N\}$ based on some rules $R$. The rule could be either of the following items. $\tilde{X}$ is the optimal choice with some evaluation indexes for classifiers; the feature subset has the lowest dimension for a given accuracy; the conditional probability distribution function for the data and that of the selected features remain the same; the error rate of the classifier would not be reduced by not increasing or decreasing the number of features. By such a selection process, we could get rid of either redundant or irrelevant features without incurring much loss of information. The distinguishing features can be found, and in this way, the dimension of data space declines, the complexity of data reduces and, especially, the performance of classification and prediction can be improved. Because of the direct interpretability of the data, feature selection is widely used in many fields such as genomics, medical image analysis, computer vision, speech recognition, computer vision, information retrieval, time series prediction [11–13], etc.

According to different ways of combining the evaluation criteria and classifiers, feature selection methods can be divided into five types, i.e., filter, wrapper, embedded, ensemble and hybrid methods [14]. Filter methods mainly depend on the attribute of features, and the evaluation criteria depend only on the original data, but not on classifiers [15]. Wrapper methods directly take the performance of the classifiers as the evaluation criterion for the selection of feature subsets; thus, the results of wrapper methods are related to specific classifiers [16]. Methods of embedding filter methods and wrapper methods are called embedded methods. For embedded methods, they are usually composed of two stages. Firstly, filter methods are used to eliminate most of the irrelevant and noise features, so as to reduce the data dimension of the subsequent search process effectively.

The second stage adopts wrapper methods to carry out the further feature selection process [17]. Ensemble methods are based on different sampling strategies to extract multiple sample sets, and then, they use a specific feature selection algorithm to obtain multiple sets of feature subsets. These feature subsets are further integrated to obtain a more stable feature subset [18]. Compared with the above three methods, the performance of the ensemble methods no longer depends merely on a single subset selected, but it is still limited since it uses only one specific feature learner. Hybrid methods can be combined with some different feature selection methods. Hybrid approaches combine two or more well-studied feature selection algorithms to form a new strategy and achieve a complementary advantage of different feature selection methods to solve a particular problem [19,20]. The hybrid approach usually capitalizes on the advantages from the sub-algorithms and therefore is more robust compared with single approaches. The feature selection techniques mentioned above have been applied to many fields of dimensionality reduction analysis [21–23]. In addition to the above five types of feature selection methods, some traditional statistical methods can also be used to reduce dimensionality, such as hypothesis testing, correlation coefficients, etc. These methods can obtain features with higher distinguishing ability, so as to improve the discriminative capacity of different classes [24,25].

Motivated by identifying biomarkers of SZ that are associated with cognitive composite ability and specific cognitive domains such as attention, working memory and verbal learning, in this paper, by proposing a hybrid feature selection method combining both machine learning and traditional statistical approaches, we explore the brain abnormalities of SZ. The data have 410 features, including both functional and structural MRI, i.e., functional network connectivity (FNC) and source-based morphometric (SBM) of 40 patients with SZ and 46 healthy controls (HCs). By applying our method to these two datasets, the results show that there exist six aberrant brain regions and 17 abnormal functional connections between the SZ group and HC group. Among our findings, there was an obvious decrease, as well as increase of both the grey matter volume and the connectivity of brain regions. The decreasing regions mainly appeared in the default mode network (DMN) and salience network (SN), e.g., the grey matter volume of precuneus (PCUN) and caudate (CAU), and the connectivity of these two brain regions, as well as insula (INS) and CAU were significantly reduced. Moreover, all connectivity corresponding with rolandic operculum and insula significantly reduced [26–31]. The significantly increased grey matter volume of brain regions was mainly distributed in frontal gyrus (FG) and supramarginal gyrus (SMG), and there also existed four with significantly increased connectivity, such as middle frontal gyrus and superior occipital gyrus, as well as middle occipital gyrus and fusiform gyrus, and the corresponding conclusion of increasing also was discussed [28,29,32]. To further confirm the significance of the selected abnormal functional connections, we also used complex network analysis. Since the level of response activity in brain regions and the ability of functional connectivity between different brain regions can reflect the degree of brain disorders, the results have the potential to provide evidence for accurate diagnosis and further for precision medicine learning of such kinds of psychiatric diseases.

## 2. Methodology

There are many feature selection methods based on machine learning, as well as traditional statistics. Combining both of them, especially developing a kind of hybrid feature selection method, is still worthy of study. In this section, we will introduce a hybrid feature selection method combining three kinds of machine learning methods and three kinds of statistical methods. In addition, some graph theory will be presented to verify the validation of the features selected by the proposed hybrid feature selection method.

*2.1. Feature Selection Methods Based on Machine Learning*

2.1.1. Feature Selection with Support Vector Machine

Support vector machine based on recursive feature elimination (SVMRFE) is a multi-variable wrapper feature selection algorithm, and it can keep relevant features and remove relatively insignificant feature variables in order to achieve higher classification performance. SVMRFE was first proposed for gene selection [33], and it has been widely applied to MRI data research, text analysis and biological information processing [34–36].

For SVMRFE, the scoring function for each feature $i$ is defined as:

$$Score(i) = |\omega_i| \quad or \quad Score(i) = \omega_i^2 \tag{1}$$

where $\omega_i$ is the weight for feature $i$ as obtained from the SVM training. Thus, features that contribute the most to discriminating the two classes are represented by $|\omega|$ with the highest values, and features with small scores are generally considered as noise, redundant or irrelevant to the problem. Therefore, eliminating features with smaller scores does not bring about great changes of the optimization problem, which is the essence of the algorithm [37,38]. The SVMRFE algorithm is briefly described as below.

---

**Algorithm 1:** Support vector machine based on recursive feature elimination (SVMRFE)
**Input:** Dataset $D$
**Process:**
    1. Initialization
Let the current feature subset $Current\_D$ contain all features, and the optimal feature subset $Best\_D = \varnothing$;
    2. Training the classifier
Train a SVM on the training set with the $Current\_D$, and evaluate the classification accuracy on the test set;
    3. Updating $Current\_D$
Calculate the importance of each feature in $Current\_D$ by the scoring function (1), and eliminate features with the smallest score;
    4. Updating $Best\_D$
If the accuracy rate of $Current\_D$ is greater than that of $Best\_D$, then let $Best\_D = Current\_D$;
    5. Repeat Steps 2–4 until the stop condition is satisfied.
**Output:** The optimal feature subset $Best\_D$

---

The stopping criterion can be a desired dimensionality, a pre-specified number of iterations or a generalization of the performances, etc.

2.1.2. Feature Selection with Random Forest

Random forest (RF) is an ensemble machine learning method using tree-type classifiers. It is built by bootstrap sampling technology and random splitting technology, and the final classification result is made by a majority vote of the trees [39,40]. Because of its excellent generalization performance, RF is also further used for feature selection [41,42].

For a given tree, let $S_0$ denote the set of input predictor data vectors and $S_j$ be the subset of the predict data reaching node $j$ in the binary split tree. According to the performance of the current feature on node $j$, $S_j$ can be divided into two subsets, i.e., $S_j^L$ and $S_j^R$; here, $S_j^L \bigcup S_j^R = S_j$ and $S_j^L \bigcap S_j^R = \varnothing$. Choosing the best split according to the mean decrease of the Gini index, which is defined as:

$$\Delta Gini_i(j) = Gini(j) - (\frac{|S_j^L|}{|S_j|} Gini(j^L) + \frac{|S_j^R|}{|S_j|} Gini(j^R)) \tag{2}$$

where $Gini(j) = 1 - \sum_{c \in C} P_c^2$ is the Gini index at node $j$. This metric reflects the contribution of each feature to node $j$; therefore, we can get an estimate of feature $i$ with Gini importance:

$$Score_{Gini}(i) = \frac{1}{n_{tree}} \sum_{t=1}^{n_{tree}} \sum_j \Delta Gini_i(j, t) \tag{3}$$

where $\Delta Gini_i(j, t)$ is the value of $\Delta Gini_i(j)$ on one tree $t$. The Gini importance indicates how large its overall discriminative value is for the classification task. We randomly chose a feature $i$, calculated its Gini importance defined in (4) and removed the features with Gini importance below feature $i$. The algorithm for feature section with random forest by Gini importance (RFFS-GI) is briefly described as below.

---

**Algorithm 2:** Feature section with random forest by Gini importance (RFFS-GI)
**Input:** Dataset $D$;
**Process:**
    1. Randomly choose a feature $i$ into the feature set;
    2. Calculate the Gini importance of all features in the feature set with the scoring function (3);
    3. Keep features with Gini importance above that of the feature $i$;
**Output:** Optimal feature subset

---

In addition, for bootstrap sampling technology, about 1/3 of the samples will not be collected at the end, and they are called the out of bag (OOB) data [43]. The role of OOB data can be considered as equivalent to the test data. Therefore, we can also use the classification accuracy of the random forest classifier on the OOB data as the feature separability criterion, so as to calculate the importance of each feature:

$$Score_{OOB}(i) = \sum \frac{ooberr2 - ooberr1}{N} \tag{4}$$

where $ooberr1$ is the classification error of the OOB data, $ooberr2$ is the classification error of the OOB data with adding noise on feature $i$ and $N$ indicates the number of trees in a random forest. We can understand that if a feature is randomly disturbed, the classification error of the OOB data will increase greatly, and it can be considered that this feature has a great influence on the classification result. The algorithm of feature section with random forest by the classification accuracy on the OOB data (RFFS-OOB) is briefly described as below.

---

**Algorithm 3:** Feature section with random forest by the classification accuracy on the OOB data (RFFS-OOB)
**Input:** Dataset $D$
**Process:**
    1. Generate random forest;
    2. Calculate feature importance based the scoring function (4), and sort the scores;
    3. The top ranked features are selected as the optimal feature subset.
**Output:** Optimal feature subset.

---

In order to improve the accuracy of feature selection results for the SBM and FNC data, we used SVMRFE, RFFS-GI and RFFS-OOB, and repeated them 20 times separately, counted the frequency of the selected features by each feature selection method and integrated the optimal feature subsets.

### 2.2. Feature Section Based on Statistical Methods

For classical statistical methods, the discriminative ability of a feature can be quantitatively measured by its contribution on distinguishing different classes [25,44].

The Kendall tau correlation coefficient provides a distribution-free test of independence between two variables. The Kendall tau correlation coefficient of feature $j$ can be defined as:

$$\tau_j = \frac{n_c - n_d}{n_1 \times n_2} \tag{5}$$

where $n_c$ and $n_d$ are the numbers of concordant and discordant pairs, respectively, and $n_1$ and $n_2$ correspond to the number of two classes of samples, respectively. For a pair of data $(x_{ij}, y_i)$ and $(x_{kj}, y_k)$ of feature $j$, it is a concordant pair when $sgn(x_{ij} - x_{kj}) = sgn(y_i - y_k)$, where $sgn()$ is the signum function (i.e., $sgn(x) = -1$ with $x < 0$, $sgn(x) = 0$ with $x = 0$ and $sgn(x) = 1$ with $x > 0$). Correspondingly, it is a discordant pair when $sgn(x_{ij} - x_{kj}) = -sgn(y_i - y_k)$. The discriminative power of each feature $j$ is defined as the absolute value of its Kendall tau correlation coefficient.

The permutation test is a non-parametric test method, which is suitable for the case of a small sample size and unknown sample distribution. Assume that there are two samples $x_A$ and $x_B$, and $\bar{x}_A$ and $\bar{x}_B$ denote the corresponding sample mean, say $n_A$ and $n_B$ are the corresponding sample size. At first, we calculate the observed test statistic $T_{obs} = \bar{x}_A - \bar{x}_B$. Then, the two samples are merged and divided into two groups with size $n_A$ and $n_B$. For each division, the difference between the mean values of the two groups is calculated and recorded. The calculated difference set is the accurate distribution of the difference under the null hypothesis. Finally, the ratio of the absolute value of the calculated difference greater than or equal to the absolute value of $T_{obs}$ is the $p$-value based on the two-sided test.

By the two-sample $t$-test, we can also determine whether there are significant differences of each feature. The $t$-value of the feature $j$ can be defined as:

$$t_j = \frac{|\bar{x}_1 - \bar{x}_2|}{\sqrt{\frac{(n_1-1)s_1^2 + (n_2-1)s_2^2}{n_1+n_2-2} \cdot \left(\frac{1}{n_1} + \frac{1}{n_2}\right)}} \tag{6}$$

where $\bar{x}_1$ and $\bar{x}_2$ are the means of feature $j$ of patients and health controls (HCs) and $s_1$ and $s_2$ represent the corresponding standard deviations. With the Kendall tau correlation coefficient, permutation test and two-sample $t$-test, we can identify features with significant differences.

### 2.3. Hybrid Feature Selection Based on Both Machine Learning and Statistical Methods

By combining the above machine learning methods and statistical methods, we propose a hybrid feature selection approach. In more detail, for machine learning methods, we summed the frequencies of SVMRFE, RFFS-GI and RFFS-OOB, then we selected the features with total frequency greater than a given value $b$ to obtain the significant feature subset. At the same time, we selected features with the absolute values of the Kendall correlation coefficient greater than a given value $c$ and those with the $p$-value of two-sample $t$-test, as well as that of the permutation test less than 0.05 as the significant feature subset, respectively. Finally, we integrated the significant feature subset from both the machine learning and statistical method as the optimal feature subset. The above process is a hybrid feature selection procedure, and the flowchart is shown in the Figure 1. The experiment results will show that the proposed hybrid feature section method is an effective attempt to combine machine learning and the statistical methods.

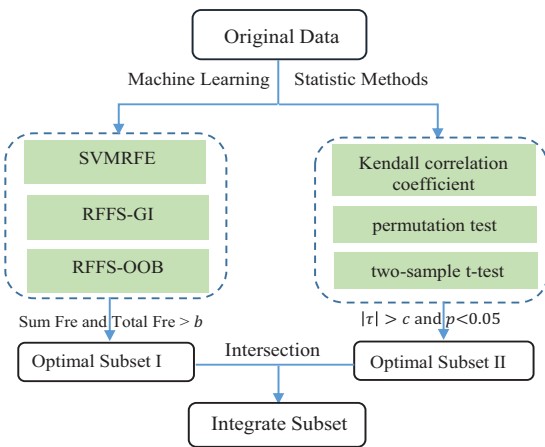

**Figure 1.** The flowchart of the hybrid feature selection method. Fre denotes frequency, $\tau$ the Kendall correlation coefficient, $p$ the $p$-values of test and $b$ and $c$ the given constants. In which, SVMRFE refers to support vector machine based on recursive feature elimination, RFFS-GI refers to the feature selection with random forest by Gini importance and RFFS-OOB refers to the feature selection with random forest by the classification accuracy on the OOB data.

### 2.4. Complex Network Analysis Based on Graph Theory

The data we used here are a type of MRI data, which contain both the regions and the functional connection information of brains. The hybrid feature selection method can be directly used to explore the disease-related abnormal brain regions and abnormal function connections. Furthermore, since the completion of various tasks allocated for brains is implemented by the coordination and cooperation between various brain regions, so it is necessary to discover the connection networks of brains in depth.

The analysis of complex network properties by several indexes (see Figure A1) can characterize the topological attributes of the network; for example, the clustering coefficient quantifies the functional segregation of the brain network, in which the functional segregation reflects the ability of a specialized process to occur within some densely-interconnected groups of the brain regions. The length of characteristic path quantifies the functional integration of the brain network, and the functional integration reflects the ability to combine rapidly some specialized information from distributed brain regions [45]. Both global and local network efficiencies quantify the transmission capability of the brain network, and the transmission capability reflects the ability of transmitting information between different brain regions in the brain network. The main difference is that the global network efficiency focuses on the global brain network, but the local network efficiency just focuses on the local brain network. Thus, by complex network analysis, we can confirm the significance of those selected abnormal connection features and can further explore the mechanism of SZ.

### 3. Experiments

In this section, based on the hybrid feature selection method and network topological analysis, we located the brain abnormalities of both regions and connections with SZ. Firstly, by the SVMRFE, RFFS-GI, RFFS-OOB, correlation coefficient and hypothesis test, the candidates of brain regions and connections associated with SZ were selected separately, and then, by the hybrid method, we could confirm the significant regions and connections of SZ. Furthermore, the complex network analysis based on graph theory was used to verify the selected abnormal connections. Ultimately, we could locate some of the abnormal brain regions and abnormal connections with SZ, which provided theoretical guidance for the rapid and accurate diagnosis of psychiatric diseases and adjuvant therapy.

*3.1. Data Collection and Preprocessing*

In this study, the Machine Learning for Signal Processing (MLSP) 2014 Schizophrenia classification challenge data were used. The data can be download from https://www.kaggle.com/c/mlsp-2014-mri. They were collected on a 3T MRI scanner at the Mind Research Network and funded by the Centers of Biomedical Research Excellence. Image preprocessing was performed using statistical parametric mapping software (SPM, http://www.fil.ion.ucl.ac.uk/spm). Further feature extraction was done by the GIFT Toolbox (http://mialab.mrn.org/software/gift/), yielding different imaging modalities, i.e., SBM and FNC features for structural MRI and resting state functional MRI, correspondingly.

The data consisted of 40 patients with SZ and 46 HCs. A diagnosis of SZ was made by using the Structured Clinical Interview for DSM-IV (SCID; Diagnostic and Statistical Manual of Mental Disorders, DSM) [46]. Each sample had 410 features (32 for SBM and 378 for FNC). SBM features were weights of brain regions, and they indicated the concentration of grey matter in different regions of the subject's brain [47]. FNC features were the pair-wise correlation values between the time-courses of 28 brain regions and can be seen as a functional modality feature describing the subjects' overall level of synchronicity between brain areas [48]. These 28 brain regions were selected according to the anatomical automatic labeling (AAL) template, and they are shown in Figure A2, while the connections between the brain regions corresponding to these FNC features are shown in Figure A3.

*3.2. Locating the Abnormalities in Brains for SZ*

For both the FNC and SBM data, we performed feature selection methods based on machine learning and statistical approaches, respectively. By the hybrid process, the key features can be selected; namely for SBM data, we obtained the abnormal brain region, and for FNC data, the abnormal connectivities were achieved. Further, we used the brain network based on graph theory to analyse the selected abnormal connections. The following Figure 2 shows the whole flowchart of the procedure.

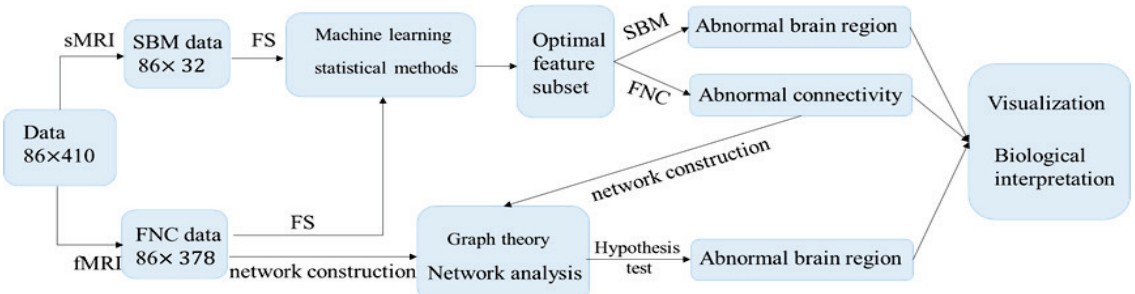

**Figure 2.** The flowchart of locating the abnormalities in brains for SZ. Where SBM refers to source-based morphometric, FNC refers to functional network connectivity and FS refers to feature selection.

### 3.2.1. Feature Selection Results Based on Machine Learning Methods

SVMRFE, RFFS-GI and RFFS-OOB were applied to perform feature selections on the MRI data respectively, with each method being repeated 20 times. Since these three methods were implemented based on the classification results and SBM data and FNC data had different classification performance, therefore, in order to obtain the key features of the two types of data more clearly, we selected the features of both of them separately. By the three feature selection methods, the results of the frequency of each feature that has been selected are shown in Figures 3 and 4 and Figures A4–A7.

It is generally believed that if the frequency of occurrence of a feature is too low, then the feature is not significant. Therefore, we only considered features with a higher frequency to obtain the significant feature subset. In Figure A8, the corresponding characteristic frequency distribution with a frequency greater than or equal to 50 is shown. Each point in this figure corresponds to the number of features with a frequency of occurrence greater than or equal to $x$. Further, we selected features with a frequency greater than or equal to 55, which is a balance between the numbers of features and the frequency

(the details can be found in the illustration of Figure A8). From Figures 3 and 4 and Figures A4 and A7, we can obtain the features of SBM data that are significant for distinguishing the HCs and SZ, and the corresponding indexes were 3, 7, 11, 24, 26, 30 and 32. We can also obtain the discriminative features of FNC data with indexes 244, 295, 183, 243, 33, 37, 40, 189, 220, 48, 78, 279, 353, 13, 185, 211, 265, 292, 328, 337 and 165.

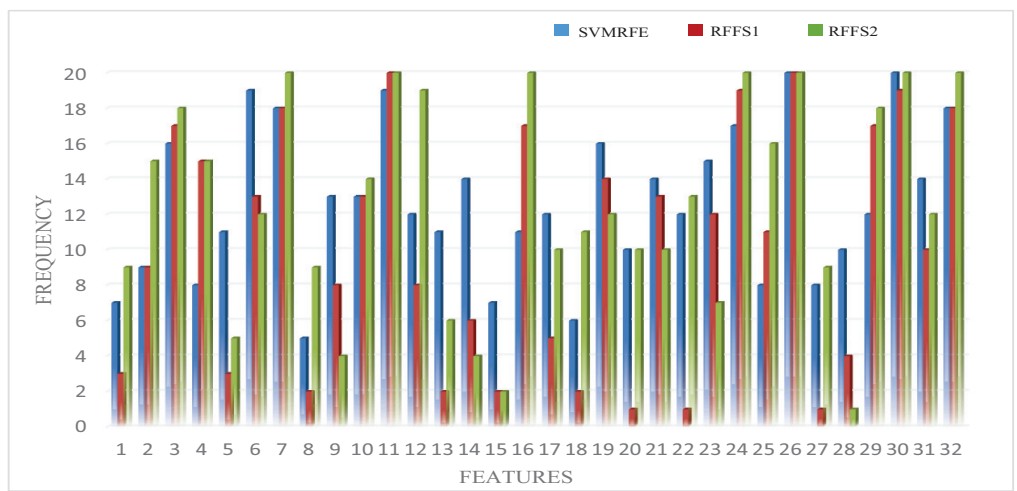

**Figure 3.** SVMRFE, RFFS-GI and RFFS-OOB results of SBM data.

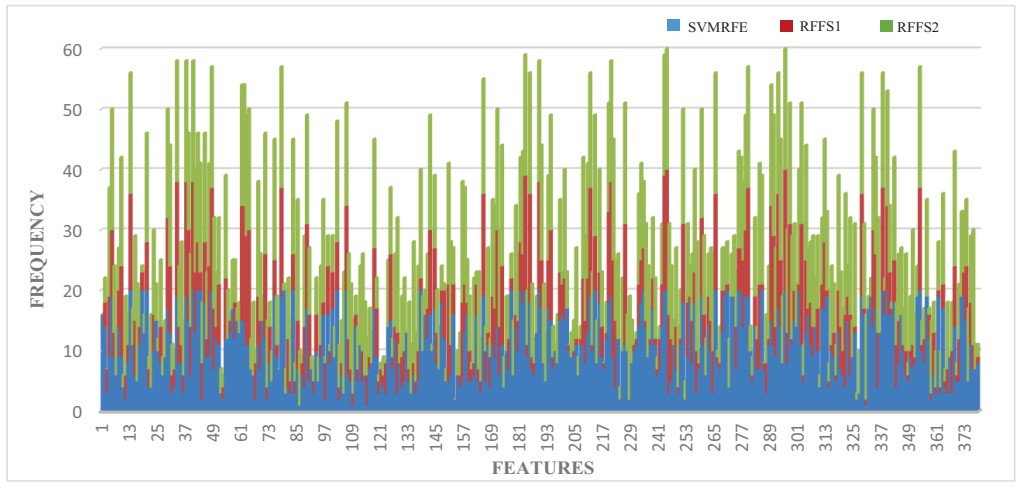

**Figure 4.** SVMRFE, RFFS-GI and RFFS-OOB results of FNC data.

### 3.2.2. Feature Selection Results Based on Statistical Methods

Statistical methods were utilized to screen out features with significant differences. The results of the Kendall correlation coefficient are shown in Figure 5, and the hypothetical test results are shown in Figure 6.

We selected features with the *p*-value of the hypothesis test less than 0.05 and the absolute value of Kendall correlation coefficient greater than 0.26, which is a balance between the size of the selected feature subsets and their distinguishing ability of SZ. The results are shown in Figure 7, where $\tau$ is the Kendall correlation coefficient and $p_1$ and $p_2$ are the *p*-values of the two-sample *t*-test and the permutation test, respectively.

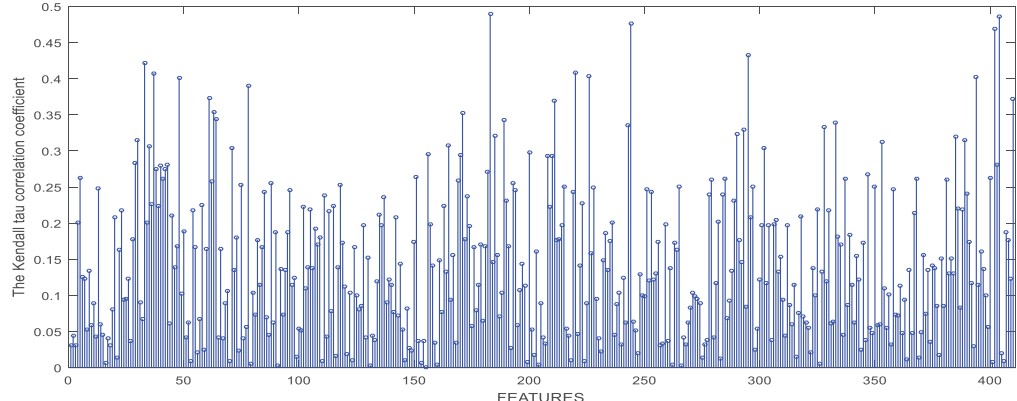

**Figure 5.** The results obtained by the Kendall correlation coefficient. The *x* axis corresponds to the features, and the *y* axis is the absolute value of the Kendall tau correlation coefficient.

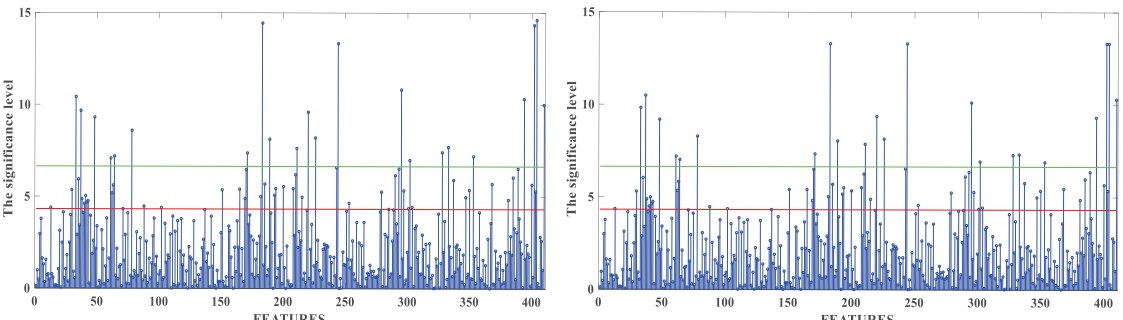

**Figure 6.** The results of hypothesis test for both two-sample *t*-tests and the permutation test. The *x* axis corresponds to the features, and the *y* axis is the significance level ($-log_2 P$). The red and green lines show the significance levels of 0.05 and 0.01, respectively. The features with $-log_2 P$ values above the lines have significant differences, and they are the candidates of abnormal regions or connections.

| SBM data | Fea \ Value | 3 | 7 | 11 | 16 | 22 | 24 | 25 | 26 | 32 |
|---|---|---|---|---|---|---|---|---|---|---|
| | $|\tau|$ | 0.26080 | 0.31956 | 0.31522 | 0.40217 | 0.26304 | 0.46848 | 0.28152 | 0.4858 | 0.37174 |
| | $p_1$ | 0.03503 | 0.01520 | 0.01092 | 0.00077 | 0.01964 | 0.00005 | 0.02552 | 0.00004 | 0.00098 |
| | $p_2$ | 0.03459 | 0.01589 | 0.01209 | 0.00160 | 0.01979 | 0.00040 | 0.02469 | 0.00040 | 0.0008 |

| FNC data | Fea \ Value | 33 | 37 | 40 | 48 | 61 | 64 | 78 | 165 | 171 | 183 | 185 | 189 |
|---|---|---|---|---|---|---|---|---|---|---|---|---|---|
| | $|\tau|$ | 0.422 | 0.408 | 0.279 | 0.401 | 0.372 | 0.344 | 0.390 | 0.308 | 0.352 | 0.489 | 0.320 | 0.342 |
| | $p_1$ | 0.001 | 0.001 | 0.041 | 0.002 | 0.007 | 0.007 | 0.003 | 0.024 | 0.006 | 0.001 | 0.019 | 0.004 |
| | $p_2$ | 0.001 | 0.001 | 0.045 | 0.002 | 0.007 | 0.008 | 0.003 | 0.023 | 0.006 | 0.001 | 0.019 | 0.004 |
| | Fea \ Value | 211 | 220 | 226 | 243 | 244 | 279 | 295 | 302 | 328 | 333 | 337 | 353 |
| | $|\tau|$ | 0.370 | 0.409 | 0.403 | 0.336 | 0.476 | 0.260 | 0.433 | 0.304 | 0.333 | 0.339 | 0.261 | 0.312 |
| | $p_1$ | 0.005 | 0.001 | 0.003 | 0.011 | 0.001 | 0.026 | 0.001 | 0.008 | 0.006 | 0.005 | 0.017 | 0.007 |
| | $p_2$ | 0.004 | 0.002 | 0.004 | 0.011 | 0.001 | 0.027 | 0.001 | 0.008 | 0.006 | 0.006 | 0.018 | 0.008 |

**Figure 7.** Feature selection results based on statistical methods.

### 3.2.3. Feature Selection Results Based on a Hybrid Method

By both machine learning and statistical methods, the key candidate features for SZ were selected, and the dataset were quite similar. We adopted the intersection of them as the final selected feature subset, and thus, the abnormal brain regions from the SBM data (see Figure 8) and the abnormal functional connectivity from the FNC data (see Figure 9) can be obtained.

| Machine learning | 3 | 7 | 11 | 24 | 26 | 30 | 32 | | |
|---|---|---|---|---|---|---|---|---|---|
| Statistical methods | 3 | 7 | 11 | 16 | 22 | 24 | 25 | 26 | 32 |
| Intersection | 3 | 7 | 11 | 24 | 26 | 32 | | | |
| Brain region (abbr.) | SMG | CG | MFG | PCUN | SFG | CAU | | | |

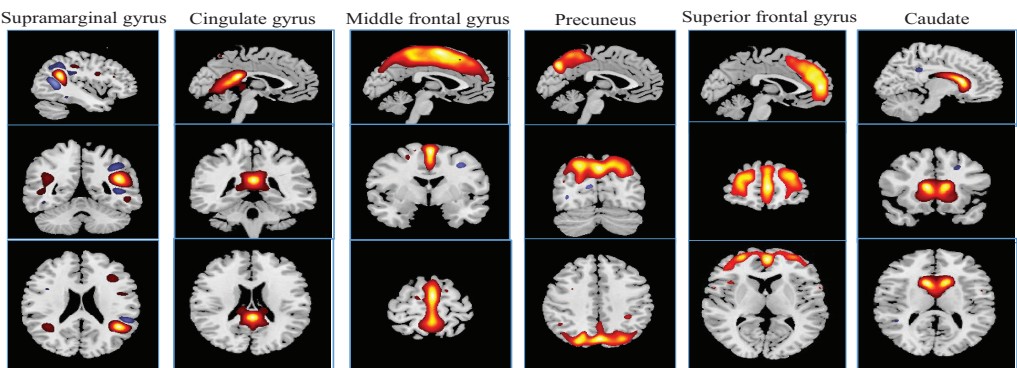

**Figure 8.** The selected abnormal brain regions of SZ by the hybrid method. Segall et al. presented the relationships between the cortical maps and the brain regions described by the SBM features [47].

Figure 8 shows the brain regions selected by our method that differed from healthy controls in SZ, and these abnormal brain regions were mainly distributed in supramarginal gyrus (SMG), cingulate gyrus (CG), middle frontal gyrus (MFG), precuneus (PCUN), superior frontal gyrus (SFG) and caudate (CAU). Compared with the HC group, the SZ group had significantly reduced grey matter volumes in the CG, PCUN and CAU and significantly increased grey matter volume of brain regions including SMG, MFG and SFG.

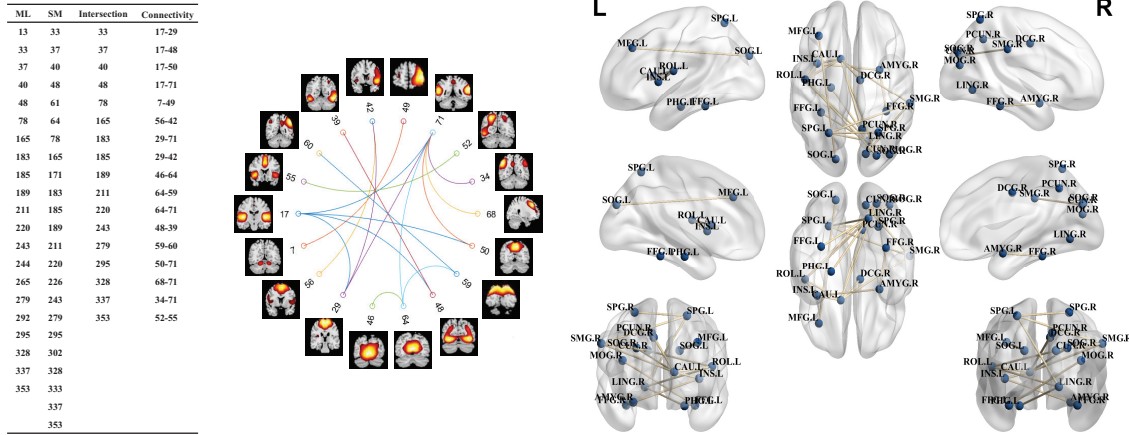

**Figure 9.** The abnormal functional connections of brains with SZ. In this figure, the left table lists the selected abnormal functional connections of the regions of interest (the relationships of the regions and the labels are shown in Figure A2), in which ML refers to machine learning methods and SM refers to statistical methods. The circular connectivity graph in the middle is a schematic map of the selected functional connections, which are listed in the fourth column of the left table. The labels in this graph correspond to the regions of interest, and the corresponding spatial maps of these regions (see [48]) are also shown in this graph. The right graph depicts the locations and their connections of the selected brain regions by the BrainNet Viewer toolbox [49].

Figure 9 shows that by the hybrid feature selection method proposed here, 17 abnormal functional connections between the SZ group and HC group can be discovered. Furthermore, by combining with the relationship between the connections and the regions shown in Figure A2, six connections are related to the caudate nucleus (CAU), including rolandic operculum (ROL), insula (INS), supramarginal gyrus (SMG), superior occipital gyrus (SOG), precuneus (PCUN) and median cingulate and paracingulate gyri. In addition, there also existed three abnormal functional connections related to the insula (i.e., ROL, amygdala and CAU) and four aberrant functional connection in rolandic operculum (i.e., insula, lingual gyrus, superior parietal gyrus and caudate). Among these abnormal connections discovered by our method, we can find that all connectivities corresponding with rolandic operculum and insula had significantly reduced, and these connectivities related to caudate nucleus had significantly decreased except the median cingulate and paracingulate gyri. Other than that, we also observed the significantly increased connectivity in middle frontal gyrus and superior occipital and middle occipital gyrus and fusiform gyrus, as well as left and right superior parietal gyrus. In conclusion, the brain connectivity in SZ generally decreased, but also had little increased connectivity. To show these abnormal connections more vividly, in Figure 9, we used the BrainNet Viewer toolbox to draw the precise locations of two brain regions with aberrant connections and to show the aberrant brain connectivity network in SZ [49] .

*3.3. Network Evaluation*

Further, to support the validity of the connectivity findings by the above hybrid feature selection method, we constructed a brain network based on these connections and explored its topological properties [50,51]. More specifically, we first chose the clustering coefficient ($C$), characteristic path length ($L$), global network efficiency ($Eg$) and local network efficiency ($Eloc$) as the evaluation index for each network. Then, we constructed weight networks with a threshold of one for the original and selected FNC data. At last, these four parameters of both SZ and HCs were calculated and tested by a two-sample *t*-test. The *p*-values of these four parameters were $1.70 \times 10^{-1}$, $5.02 \times 10^{-3}$, $2.99 \times 10^{-2}$ and $4.27 \times 10^{-2}$ for the original FNC data and $6.64 \times 10^{-2}$, $3.41 \times 10^{-6}$, $5.40 \times 10^{-6}$ and $1.90 \times 10^{-2}$ after feature selection by our method. Obviously, from the results of the *p*-values of the four parameters, we can find that the *p*-values of all these parameters decreased significantly after feature selection, which means that the distinction of four parameters between the HCs and SZ became more apparent after feature selection, especially the characteristic path length and the global network efficiency. This shows that the HCs and SZ become obviously distinguishable by the hybrid feature selection method and shows the validity of our method.

## 4. Discussion

The methods based on machine learning pay more attention to the classification accuracy, but the statistical methods emphasize the correlation between feature and label, which explains the essential difference between the two approaches. Comparing the significant subsets selected by these two approaches, it is clear that most of the biomarkers in these two subsets were same, and this means that despite the emphasis of the two approaches being different, both of them did find the significant features. Further, by integrating the significant subset of these two approaches, the significant features can be double checked and obtained finally by the hybrid method proposed in this article. For example, for the data before feature selection, the *p*-value of characteristic path length, which is referred to as $L$ in the above section, was $5.02 \times 10^{-3}$. The *p*-value of $L$ for the optimal subset I, which was obtained by machine learning methods, the *p*-value of $L$ for the optimal subset II, which was obtained by statistic methods, and the *p*-value of $L$ for the optimal subset by the proposed hybrid method were $9.40 \times 10^{-6}$, $2.82 \times 10^{-5}$ and $3.41 \times 10^{-6}$ respectively. The results show that the HCs and SZ became obviously distinguishable after feature selection; specially, our method was more significant than machine learning, as well as statistical methods. In summary, the hybrid method can combine the

strength of both machine learning and statistic methods to improve the accuracy of the results, and the results of network evaluation also confirmed this point.

Our findings are quite consistent with those reports that the grey matter volume of CG, PCUN and CAU is significantly reduced in SZ [26,27,52,53]. The CG is considered to be a brain region closely related to task attention, memory and affection, which has been reported to be destroyed in SZ [54]. The PCUN is the portion of the superior parietal lobule on the medial surface of each brain hemisphere, and it is often considered to be a brain region that plays an important role in the pathogenesis of SZ [55]. Given that the Behavioural Inhibition System (BIS) activity and Cloninger's Temperamental Dimension Harm Avoidance (HA) are mainly bound up with the study of the anxiety trait [56,57] and the research results show that the BIS-sensitively as well as HA are negatively correlated with the regional gray matter volume at the CG and PCUN, the SZ may be accompanied by anxiety trait due to the reduction of the gray matter volume at these two regions [58]. The CAU is one of the structures that makes up the dorsal striatum, which is a component of the basal ganglia. It can affect the cognitive function of patients, resulting in decreased memory ability, and may be the cause of cognitive dysfunction in SZ [59].

In our findings, most of the brain connectivity in SZ was significantly reduced, which had been generally accepted as the fact that the functional connectivity reduces significantly in SZ and the reduction may cause the damage of information integration [60]. Among these abnormal connectivities, CAU, INS and ROL were the most connected regions. The INS mainly participates in the formation of aversion, the regulation of pain, the production of depression, the regulation of cardiac activity and the planning of language [61], and these may be the cause of affective symptoms in SZ. Moreover, many studies have found that the connectivity in the INS decreased, which may cause the disrupted functional integration of the brain [30]. The ROL is mainly involved in language, and Wu et al. suggested that the reduction of connectivity of ROL improves the vulnerability of speech recognition to speech masking [62]. Not only that, the work also showed that the ROL is bound up with hallucination [63]. It has been reported that SZ is often accompanied by motor abnormalities, and the work showed that the abnormalities of the motor system are related to the abnormal functional connectivity of CAU and CG [64]. In addition, the work showed that the network of DMN including posterior cingulate cortex and lateral temporal cortex and SN including INS and CAU have abnormal connectivity in SZ [65]. DMN is mainly related to oriented attention and self-monitoring [66], and SN is implicated in orienting toward salient external stimuli and internal events [67]. These state clearly that the abnormal connectivity of CAU and INS may result in the cognitive deficits.

In addition to the above findings that there exist some decreasing regions and connections, we also found that there exist some increasing regions in SMG, MFG and SFG and the increasing connectivity of MFG and superior occipital gyrus, the median cingulate and paracingulate gyri and CAU, the left and right superior parietal, as well as middle occipital gyrus and fusiform gyrus. Some corresponding conclusions were also mentioned in literatures [28,29,32]. Research showed that the connectivity of the frontoparietal network (FPN) and DMN significantly increased [65]. The FPN including dorsolateral prefrontal cortex and dorsolateral parietal cortex is implicated in executive control [68], which means the function of executive control of SZ is different from HCs. In conclusion, we found that most abnormal brain regions and connectivity discovered by our method were mainly related to cognition and hallucination. These abnormalities may be the reason for the cognitive deficits and autistic thinking in SZ. Moreover, our studies show that compared with HCs, the brain network of SZ is not a single decline or rise, but a mix of both. The most abnormal connectivity may cause the information integration and transmission damage. Thus, by our method, we did find the abnormal regions and the connectivity of brain that were strongly related to SZ, and the results also supported the effectiveness of using functional disconnectivity from neuroimaging as a biomarker for diagnosis of mental disorders [69].

## 5. Conclusions

By the proposed hybrid feature selection approach, which combined both machine learning and traditional statistical methods, the abnormal brain regions and abnormal connections in brains of SZ were discovered. The results of SBM data showed that the abnormal brain regions of SZ were mainly distributed in supramarginal gyrus, cingulate gyrus, middle frontal gyrus, superior frontal gyrus, precuneus and caudate. These brain regions are reported to have strong association with SZ, and they are mainly involved in perception, thinking, emotion and spiritual activity. The results of FNC data showed that most of the abnormal functional connections in brains of SZ were related to FPN, DMN and SN. These three networks are closely related to cognitive deficits, especially in executive control and salience processing. All of the results suggest that the brain regions and connectivity in SZ are destroyed compared with HCs, and the abnormal activity may cause the cognitive deficits and autistic thinking in SZ. In addition, the complex network analysis further verified the significance of the selected abnormal functional connections. All findings supported the validation of the proposed hybrid feature selection method, and thus, it is promised that such a hybrid feature selection method can be further used for other kinds of medical data analysis to enhance the diagnosis ability.

**Author Contributions:** Conceptualization, C.Q.; methodology, C.Q. and L.L.; writing, original draft preparation, L.L. and L.Y.; writing, review and editing, C.Q. and P.J.K.; funding acquisition, C.Q.

**Funding:** This research was funded by NSFC Nos. 11471006 and 11101327, the Fundamental Research Funds for the Central Universities (No. xjj2017126), the Science and Technology Project of Xi'an (No. 201809164CX5JC6) and the HPC Platform of Xi'an Jiaotong University.

**Conflicts of Interest:** The authors declare no conflict of interest.

## Appendix A

| | |
|---|---|
| Clustering coefficient (C):  Average nodal clustering coefficient. | $C = \frac{1}{n}\sum_{i \in N} C_i = \frac{1}{n}\sum_{i \in N} \frac{2t_i}{k_i(k_i-1)}$ |
| Characteristic path length (L):  integration and global routing efficiency of a network | $\frac{1}{L} = \frac{1}{n(n-1)}\sum_{i \neq j \in N} \frac{1}{d_{ij}} = E_{global}$ |
| Gamma ($\gamma$): the normalized clustering coefficient | $\gamma = C/C_{rand}$ |
| Lambda ($\lambda$): the normalized characteristic path length | $\lambda = L/L_{rand}$ |
| Sigma ($\sigma$): the extent of small-world property | $\sigma = \gamma/\lambda$ |
| Degree: number of links connected directly to a node | $k_i = \sum_{j \in N} a_{ij} \;/\; k_i = \sum_{j \in N} w_{ij}$ |
| Nodal clustering coefficient:  local clustering and closeness of neighborhood of node | $C_i$ |
| Nodal efficiency:  efficiency for a node communicating with the other | $e_i = \frac{1}{n-1}\sum_{j \in N, j \neq i} \frac{1}{d_{ij}}$ |
| Betweenness centrality:  a centrality measure in the communications between other nodes | $b_i = \frac{1}{(n-1)(n-2)}\sum_{j \neq i \neq k} \frac{\sigma_{jk}(i)}{\sigma_{jk}}$ |

**Figure A1.** Different measuring parameters of the global and local network properties. Where $t_i$ is the number of triangles around node $i$, $d_{ij}$ is the shortest path length between node $i$ and node $j$, $C_{rand}$ and $L_{rand}$ refer to the average clustering coefficient and characteristic path length values obtained from 100 random networks with the same number of nodes, as well as edges and the same degree of distribution as the original network, $\sigma_{jk}$ is the number of shortest paths between $j$ and $k$ and $\sigma_{jk}(i)$ is the number of shortest paths between $j$ and $k$ that pass through $i$.

| Labels | Regions | abbr. | x( mm) | y( mm) | z( mm) |
|--------|---------|-------|--------|--------|--------|
| 21 | Olfactory cortex | OLF.L | -8.06 | 15.05 | -11.46 |
| 17 | Rolandic operculum | ROL.L | -47.16 | -8.48 | 13.95 |
| 7 | Middle frontal gyrus | MFG.L | -33.43 | 32.73 | 35.46 |
| 23 | Superior frontal gyrus, medial | SFGmed.L | -4.8 | 49.17 | 30.89 |
| 24 | Superior frontal gyrus, medial | SFGmed.R | 9.1 | 50.84 | 30.22 |
| 38 | Hippocampus | HIP.R | 29.23 | -19.78 | -10.33 |
| 56 | Fusiform gyrus | FFG.R | 33.97 | -39.1 | -20.18 |
| 29 | Insula | INS.L | -35.13 | 6.65 | 3.44 |
| 46 | Cuneus | CUN.R | 13.51 | -79.36 | 28.23 |
| 64 | Supramarginal gyrus | SMG.R | 57.61 | -31.5 | 34.48 |
| 67 | Precuneus | PCUN.L | -7.24 | -56.07 | 48.01 |
| 48 | Lingual gyrus | LING.R | 16.29 | -66.93 | -3.87 |
| 39 | Parahippocampal gyrus | PHG.L | -21.17 | -15.95 | -20.7 |
| 59 | Superior parietal gyrus | SPG.L | -23.45 | -59.56 | 58.96 |
| 50 | Superior occipital gyrus | SOG.R | 24.29 | -80.85 | 30.59 |
| 53 | Inferior occipital gyrus | IOG.L | -36.36 | -78.29 | -7.84 |
| 25 | Superior frontal gyrus, medial orbital | ORBsupmed.L | -5.17 | 54.06 | -7.4 |
| 68 | Precuneus | PCUN.R | 9.98 | -56.05 | 43.77 |
| 34 | Median cingulate and paracingulate gyri | DCG.R | 8.02 | -8.83 | 39.79 |
| 60 | Superior parietal gyrus | SPG.R | 26.11 | -59.18 | 62.06 |
| 52 | Middle occipital gyrus | MOG.R | 37.39 | -79.7 | 19.42 |
| 72 | Caudate nucleus | CAU.R | 14.84 | 12.07 | 9.42 |
| 71 | Caudate nucleus | CAU.L | -11.46 | 11 | 9.24 |
| 55 | Fusiform gyrus | FFG.L | -31.16 | -40.3 | -20.23 |
| 42 | Amygdala | AMYG.R | 27.32 | 0.64 | -17.5 |
| 20 | Supplementary motor area | SMA.R | 8.62 | 0.17 | 61.85 |
| 47 | Lingual gyrus | LING.L | -14.62 | -67.56 | -4.63 |
| 49 | Superior occipital gyrus | SOG.L | -16.54 | -84.26 | 28.17 |

**Figure A2.** Twenty eight brain regions selected for the experiment according to the AAL template.

| Fea | R1 | R2 | Fea | R1 | R2 | Fea | R1 | R2 | Fea | R1 | R2 | Fea | R1 | R2 | Fea | R1 | R2 | Fea | R1 | R2 | Fea | R1 | R2 |
|---|---|---|---|---|---|---|---|---|---|---|---|---|---|---|---|---|---|---|---|---|---|---|---|
| 1 | 21 | 17 | 49 | 17 | 55 | 97 | 23 | 71 | 145 | 38 | 20 | 193 | 46 | 59 | 241 | 67 | 47 | 289 | 50 | 25 | 337 | 34 | 71 |
| 2 | 21 | 7 | 50 | 17 | 42 | 98 | 23 | 55 | 146 | 38 | 47 | 194 | 46 | 50 | 242 | 67 | 49 | 290 | 50 | 68 | 338 | 34 | 55 |
| 3 | 21 | 23 | 51 | 17 | 20 | 99 | 23 | 42 | 147 | 38 | 49 | 195 | 46 | 53 | 243 | 48 | 39 | 291 | 50 | 34 | 339 | 34 | 42 |
| 4 | 21 | 24 | 52 | 17 | 47 | 100 | 23 | 20 | 148 | 56 | 29 | 196 | 46 | 25 | 244 | 48 | 59 | 292 | 50 | 60 | 340 | 34 | 20 |
| 5 | 21 | 38 | 53 | 17 | 49 | 101 | 23 | 47 | 149 | 56 | 46 | 197 | 46 | 68 | 245 | 48 | 50 | 293 | 50 | 52 | 341 | 34 | 47 |
| 6 | 21 | 56 | 54 | 7 | 23 | 102 | 23 | 49 | 150 | 56 | 64 | 198 | 46 | 34 | 246 | 48 | 53 | 294 | 50 | 72 | 342 | 34 | 49 |
| 7 | 21 | 29 | 55 | 7 | 24 | 103 | 24 | 38 | 151 | 56 | 67 | 199 | 46 | 60 | 247 | 48 | 25 | 295 | 50 | 71 | 343 | 60 | 52 |
| 8 | 21 | 46 | 56 | 7 | 38 | 104 | 24 | 56 | 152 | 56 | 48 | 200 | 46 | 52 | 248 | 48 | 68 | 296 | 50 | 55 | 344 | 60 | 72 |
| 9 | 21 | 64 | 57 | 7 | 56 | 105 | 24 | 29 | 153 | 56 | 39 | 201 | 46 | 72 | 249 | 48 | 34 | 297 | 50 | 42 | 345 | 60 | 71 |
| 10 | 21 | 67 | 58 | 7 | 29 | 106 | 24 | 46 | 154 | 56 | 59 | 202 | 46 | 71 | 250 | 48 | 60 | 298 | 50 | 20 | 346 | 60 | 55 |
| 11 | 21 | 48 | 59 | 7 | 46 | 107 | 24 | 64 | 155 | 56 | 50 | 203 | 46 | 55 | 251 | 48 | 52 | 299 | 50 | 47 | 347 | 60 | 42 |
| 12 | 21 | 39 | 60 | 7 | 64 | 108 | 24 | 67 | 156 | 56 | 53 | 204 | 46 | 42 | 252 | 48 | 72 | 300 | 50 | 49 | 348 | 60 | 20 |
| 13 | 21 | 59 | 61 | 7 | 67 | 109 | 24 | 48 | 157 | 56 | 25 | 205 | 46 | 20 | 253 | 48 | 71 | 301 | 53 | 25 | 349 | 60 | 47 |
| 14 | 21 | 50 | 62 | 7 | 48 | 110 | 24 | 39 | 158 | 56 | 68 | 206 | 46 | 47 | 254 | 48 | 55 | 302 | 53 | 68 | 350 | 60 | 49 |
| 15 | 21 | 53 | 63 | 7 | 39 | 111 | 24 | 59 | 159 | 56 | 34 | 207 | 46 | 49 | 255 | 48 | 42 | 303 | 53 | 34 | 351 | 52 | 72 |
| 16 | 21 | 25 | 64 | 7 | 59 | 112 | 24 | 50 | 160 | 56 | 60 | 208 | 64 | 67 | 256 | 48 | 20 | 304 | 53 | 60 | 352 | 52 | 71 |
| 17 | 21 | 68 | 65 | 7 | 50 | 113 | 24 | 53 | 161 | 56 | 52 | 209 | 64 | 48 | 257 | 48 | 47 | 305 | 53 | 52 | 353 | 52 | 55 |
| 18 | 21 | 34 | 66 | 7 | 53 | 114 | 24 | 25 | 162 | 56 | 72 | 210 | 64 | 39 | 258 | 48 | 49 | 306 | 53 | 72 | 354 | 52 | 42 |
| 19 | 21 | 60 | 67 | 7 | 25 | 115 | 24 | 68 | 163 | 56 | 71 | 211 | 64 | 59 | 259 | 39 | 59 | 307 | 53 | 71 | 355 | 52 | 20 |
| 20 | 21 | 52 | 68 | 7 | 68 | 116 | 24 | 34 | 164 | 56 | 55 | 212 | 64 | 50 | 260 | 39 | 50 | 308 | 53 | 55 | 356 | 52 | 47 |
| 21 | 21 | 72 | 69 | 7 | 34 | 117 | 24 | 60 | 165 | 56 | 42 | 213 | 64 | 53 | 261 | 39 | 53 | 309 | 53 | 42 | 357 | 52 | 49 |
| 22 | 21 | 71 | 70 | 7 | 60 | 118 | 24 | 52 | 166 | 56 | 20 | 214 | 64 | 25 | 262 | 39 | 25 | 310 | 53 | 20 | 358 | 72 | 71 |
| 23 | 21 | 55 | 71 | 7 | 52 | 119 | 24 | 72 | 167 | 56 | 47 | 215 | 64 | 68 | 263 | 39 | 68 | 311 | 53 | 47 | 359 | 72 | 55 |
| 24 | 21 | 42 | 72 | 7 | 72 | 120 | 24 | 71 | 168 | 56 | 49 | 216 | 64 | 34 | 264 | 39 | 34 | 312 | 53 | 49 | 360 | 72 | 42 |
| 25 | 21 | 20 | 73 | 7 | 71 | 121 | 24 | 55 | 169 | 29 | 46 | 217 | 64 | 60 | 265 | 39 | 60 | 313 | 25 | 68 | 361 | 72 | 20 |
| 26 | 21 | 47 | 74 | 7 | 55 | 122 | 24 | 42 | 170 | 29 | 64 | 218 | 64 | 52 | 266 | 39 | 52 | 314 | 25 | 34 | 362 | 72 | 47 |
| 27 | 21 | 49 | 75 | 7 | 42 | 123 | 24 | 20 | 171 | 29 | 67 | 219 | 64 | 72 | 267 | 39 | 72 | 315 | 25 | 60 | 363 | 72 | 49 |
| 28 | 17 | 7 | 76 | 7 | 20 | 124 | 24 | 47 | 172 | 29 | 48 | 220 | 64 | 71 | 268 | 39 | 71 | 316 | 25 | 52 | 364 | 71 | 55 |
| 29 | 17 | 23 | 77 | 7 | 47 | 125 | 24 | 49 | 173 | 29 | 39 | 221 | 64 | 55 | 269 | 39 | 55 | 317 | 25 | 72 | 365 | 71 | 42 |
| 30 | 17 | 24 | 78 | 7 | 49 | 126 | 38 | 56 | 174 | 29 | 59 | 222 | 64 | 42 | 270 | 39 | 42 | 318 | 25 | 71 | 366 | 71 | 20 |
| 31 | 17 | 38 | 79 | 23 | 24 | 127 | 38 | 29 | 175 | 29 | 50 | 223 | 64 | 20 | 271 | 39 | 20 | 319 | 25 | 55 | 367 | 71 | 47 |
| 32 | 17 | 56 | 80 | 23 | 38 | 128 | 38 | 46 | 176 | 29 | 53 | 224 | 64 | 47 | 272 | 39 | 47 | 320 | 25 | 42 | 368 | 71 | 49 |
| 33 | 17 | 29 | 81 | 23 | 56 | 129 | 38 | 64 | 177 | 29 | 25 | 225 | 64 | 49 | 273 | 39 | 49 | 321 | 25 | 20 | 369 | 55 | 42 |
| 34 | 17 | 46 | 82 | 23 | 29 | 130 | 38 | 67 | 178 | 29 | 68 | 226 | 67 | 48 | 274 | 59 | 50 | 322 | 25 | 47 | 370 | 55 | 20 |
| 35 | 17 | 64 | 83 | 23 | 46 | 131 | 38 | 48 | 179 | 29 | 34 | 227 | 67 | 39 | 275 | 59 | 53 | 323 | 25 | 49 | 371 | 55 | 47 |
| 36 | 17 | 67 | 84 | 23 | 64 | 132 | 38 | 39 | 180 | 29 | 60 | 228 | 67 | 59 | 276 | 59 | 25 | 324 | 68 | 34 | 372 | 55 | 49 |
| 37 | 17 | 48 | 85 | 23 | 67 | 133 | 38 | 59 | 181 | 29 | 52 | 229 | 67 | 50 | 277 | 59 | 68 | 325 | 68 | 60 | 373 | 42 | 20 |
| 38 | 17 | 39 | 86 | 23 | 48 | 134 | 38 | 50 | 182 | 29 | 72 | 230 | 67 | 53 | 278 | 59 | 34 | 326 | 68 | 52 | 374 | 42 | 47 |
| 39 | 17 | 59 | 87 | 23 | 39 | 135 | 38 | 53 | 183 | 29 | 71 | 231 | 67 | 25 | 279 | 59 | 60 | 327 | 68 | 72 | 375 | 42 | 49 |
| 40 | 17 | 50 | 88 | 23 | 59 | 136 | 38 | 25 | 184 | 29 | 55 | 232 | 67 | 68 | 280 | 59 | 52 | 328 | 68 | 71 | 376 | 20 | 47 |
| 41 | 17 | 53 | 89 | 23 | 50 | 137 | 38 | 68 | 185 | 29 | 42 | 233 | 67 | 34 | 281 | 59 | 72 | 329 | 68 | 55 | 377 | 20 | 49 |
| 42 | 17 | 25 | 90 | 23 | 53 | 138 | 38 | 34 | 186 | 29 | 20 | 234 | 67 | 60 | 282 | 59 | 71 | 330 | 68 | 42 | 378 | 47 | 49 |
| 43 | 17 | 68 | 91 | 23 | 25 | 139 | 38 | 60 | 187 | 29 | 47 | 235 | 67 | 52 | 283 | 59 | 55 | 331 | 68 | 20 |  |  |  |
| 44 | 17 | 34 | 92 | 23 | 68 | 140 | 38 | 52 | 188 | 29 | 49 | 236 | 67 | 72 | 284 | 59 | 42 | 332 | 68 | 47 |  |  |  |
| 45 | 17 | 60 | 93 | 23 | 34 | 141 | 38 | 72 | 189 | 46 | 64 | 237 | 67 | 71 | 285 | 59 | 20 | 333 | 68 | 49 |  |  |  |
| 46 | 17 | 52 | 94 | 23 | 60 | 142 | 38 | 71 | 190 | 46 | 67 | 238 | 67 | 55 | 286 | 59 | 47 | 334 | 34 | 60 |  |  |  |
| 47 | 17 | 72 | 95 | 23 | 52 | 143 | 38 | 55 | 191 | 46 | 48 | 239 | 67 | 42 | 287 | 59 | 49 | 335 | 34 | 52 |  |  |  |
| 48 | 17 | 71 | 96 | 23 | 72 | 144 | 38 | 42 | 192 | 46 | 39 | 240 | 67 | 20 | 288 | 50 | 53 | 336 | 34 | 72 |  |  |  |

**Figure A3.** The connections between the brain regions R1 and R2 corresponding to FNC features.

| SVMRFE | | RFFS1 | | RFFS2 | | SVMRFE | | RFFS1 | | RFFS2 | |
|---|---|---|---|---|---|---|---|---|---|---|---|
| Fea | Fre | Fea | Fre | Fea | Fre | Fea | Fre | Fea | Fre | Fea | Fre |
| 26 | 20 | 11 | 20 | 7 | 20 | 17 | 12 | 25 | 11 | 19 | 12 |
| 30 | 20 | 26 | 20 | 11 | 20 | 22 | 12 | 31 | 10 | 31 | 12 |
| 6 | 19 | 24 | 19 | 16 | 20 | 29 | 12 | 9 | 8 | 18 | 11 |
| 11 | 19 | 30 | 19 | 24 | 20 | 5 | 11 | 12 | 8 | 17 | 10 |
| 7 | 18 | 7 | 18 | 26 | 20 | 13 | 11 | 14 | 6 | 20 | 10 |
| 32 | 18 | 32 | 18 | 30 | 20 | 16 | 11 | 17 | 5 | 21 | 10 |
| 24 | 17 | 16 | 17 | 32 | 20 | 20 | 10 | 28 | 4 | 1 | 9 |
| 19 | 16 | 29 | 17 | 12 | 19 | 28 | 10 | 1 | 3 | 8 | 9 |
| 3 | 16 | 3 | 17 | 29 | 18 | 2 | 9 | 5 | 3 | 27 | 9 |
| 23 | 15 | 2 | 16 | 3 | 18 | 4 | 8 | 8 | 2 | 23 | 7 |
| 14 | 14 | 4 | 15 | 25 | 16 | 25 | 8 | 13 | 2 | 13 | 6 |
| 21 | 14 | 19 | 14 | 2 | 15 | 27 | 8 | 15 | 2 | 5 | 5 |
| 31 | 14 | 10 | 13 | 4 | 15 | 1 | 7 | 18 | 2 | 9 | 4 |
| 9 | 13 | 6 | 13 | 10 | 14 | 15 | 7 | 20 | 1 | 14 | 4 |
| 10 | 13 | 21 | 13 | 22 | 13 | 18 | 6 | 22 | 1 | 15 | 2 |
| 12 | 12 | 23 | 12 | 6 | 12 | 8 | 5 | 27 | 1 | 28 | 1 |

**Figure A4.** SVMRFE and RFFS results of SBM data, where Fea represents the feature number and Fre represents the frequency at which the feature appears in 20 experiments.

| SVMRFE | | RFFS1 | | RFFS2 | | SVMRFE | | RFFS1 | | RFFS2 | | SVMRFE | | RFFS1 | | RFFS2 | |
|---|---|---|---|---|---|---|---|---|---|---|---|---|---|---|---|---|---|
| Fea | Fre | Fea | Fre | Fea | Fre | Fea | Fre | Fea | Fre | Fea | Fre | Fea | Fre | Fea | Fre | Fea | Fre |
| 13 | 20 | 244 | 20 | 5 | 20 | 278 | 19 | 71 | 14 | 321 | 20 | 152 | 16 | 42 | 8 | 47 | 17 |
| 18 | 20 | 295 | 20 | 13 | 20 | 279 | 19 | 89 | 14 | 328 | 20 | 158 | 16 | 68 | 8 | 54 | 17 |
| 20 | 20 | 226 | 20 | 30 | 20 | 313 | 19 | 106 | 14 | 333 | 20 | 162 | 16 | 69 | 8 | 96 | 17 |
| 40 | 20 | 302 | 20 | 33 | 20 | 328 | 19 | 142 | 14 | 353 | 20 | 199 | 16 | 81 | 8 | 106 | 17 |
| 42 | 20 | 37 | 19 | 37 | 20 | 332 | 19 | 156 | 13 | 68 | 19 | 305 | 16 | 85 | 8 | 151 | 17 |
| 43 | 20 | 183 | 19 | 40 | 20 | 352 | 19 | 173 | 13 | 83 | 19 | 329 | 16 | 98 | 8 | 170 | 17 |
| 48 | 20 | 220 | 19 | 48 | 20 | 356 | 19 | 245 | 13 | 128 | 19 | 338 | 16 | 102 | 8 | 182 | 17 |
| 79 | 20 | 243 | 19 | 61 | 20 | 361 | 19 | 251 | 13 | 142 | 19 | 340 | 16 | 119 | 8 | 215 | 17 |
| 83 | 20 | 33 | 19 | 62 | 20 | 371 | 19 | 259 | 13 | 165 | 19 | 341 | 16 | 148 | 8 | 228 | 17 |
| 102 | 20 | 289 | 19 | 63 | 20 | 372 | 19 | 304 | 12 | 190 | 19 | 345 | 16 | 157 | 8 | 256 | 17 |
| 106 | 20 | 61 | 19 | 64 | 20 | 45 | 18 | 321 | 12 | 194 | 19 | 24 | 15 | 169 | 8 | 296 | 17 |
| 138 | 20 | 292 | 19 | 71 | 20 | 147 | 18 | 5 | 11 | 210 | 19 | 38 | 15 | 170 | 8 | 298 | 17 |
| 177 | 20 | 62 | 19 | 75 | 20 | 182 | 18 | 30 | 11 | 211 | 19 | 59 | 15 | 172 | 8 | 312 | 17 |
| 181 | 20 | 64 | 19 | 78 | 20 | 185 | 18 | 90 | 11 | 251 | 19 | 61 | 15 | 181 | 8 | 325 | 17 |
| 183 | 20 | 189 | 18 | 85 | 20 | 215 | 18 | 125 | 11 | 278 | 19 | 62 | 15 | 182 | 8 | 342 | 17 |
| 189 | 20 | 297 | 18 | 102 | 20 | 219 | 18 | 208 | 11 | 301 | 19 | 70 | 15 | 190 | 8 | 16 | 16 |
| 213 | 20 | 63 | 18 | 105 | 20 | 233 | 18 | 222 | 11 | 304 | 19 | 125 | 15 | 193 | 8 | 38 | 16 |
| 234 | 20 | 78 | 18 | 135 | 20 | 251 | 18 | 277 | 11 | 318 | 19 | 194 | 15 | 198 | 8 | 113 | 16 |
| 243 | 20 | 185 | 18 | 150 | 20 | 254 | 18 | 278 | 11 | 335 | 19 | 198 | 15 | 200 | 7 | 157 | 16 |
| 244 | 20 | 290 | 18 | 156 | 20 | 256 | 18 | 293 | 11 | 337 | 19 | 200 | 15 | 209 | 7 | 163 | 16 |
| 265 | 20 | 279 | 18 | 171 | 20 | 342 | 18 | 312 | 11 | 339 | 19 | 289 | 15 | 232 | 7 | 169 | 16 |
| 270 | 20 | 211 | 18 | 173 | 20 | 29 | 17 | 323 | 10 | 350 | 19 | 299 | 15 | 233 | 7 | 172 | 16 |
| 275 | 20 | 40 | 18 | 183 | 20 | 57 | 17 | 368 | 10 | 368 | 19 | 322 | 15 | 234 | 7 | 233 | 16 |
| 276 | 20 | 328 | 17 | 185 | 20 | 89 | 17 | 4 | 10 | 376 | 19 | 373 | 15 | 242 | 7 | 235 | 16 |
| 285 | 20 | 353 | 17 | 189 | 20 | 100 | 17 | 41 | 10 | 4 | 18 | 2 | 14 | 246 | 7 | 264 | 16 |
| 295 | 20 | 165 | 17 | 208 | 20 | 118 | 17 | 45 | 10 | 9 | 18 | 8 | 14 | 248 | 7 | 275 | 16 |
| 301 | 20 | 337 | 17 | 213 | 20 | 144 | 17 | 49 | 10 | 20 | 18 | 110 | 14 | 255 | 7 | 323 | 16 |
| 337 | 20 | 48 | 17 | 220 | 20 | 190 | 17 | 51 | 10 | 29 | 18 | 124 | 14 | 257 | 7 | 334 | 16 |
| 339 | 20 | 150 | 16 | 221 | 20 | 193 | 17 | 54 | 10 | 42 | 18 | 171 | 14 | 264 | 7 | 347 | 16 |
| 353 | 20 | 171 | 16 | 226 | 20 | 201 | 17 | 88 | 10 | 43 | 18 | 179 | 14 | 268 | 7 | 356 | 16 |
| 5 | 19 | 265 | 16 | 243 | 20 | 238 | 17 | 118 | 10 | 45 | 18 | 232 | 14 | 274 | 7 | 49 | 15 |
| 33 | 19 | 333 | 16 | 244 | 20 | 292 | 17 | 128 | 10 | 89 | 18 | 235 | 14 | 275 | 7 | 88 | 15 |
| 37 | 19 | 13 | 16 | 265 | 20 | 311 | 17 | 144 | 10 | 118 | 18 | 242 | 14 | 276 | 7 | 205 | 15 |
| 75 | 19 | 221 | 16 | 279 | 20 | 312 | 17 | 210 | 9 | 138 | 18 | 262 | 14 | 306 | 7 | 232 | 15 |
| 78 | 19 | 9 | 15 | 284 | 20 | 331 | 17 | 213 | 9 | 179 | 18 | 277 | 14 | 309 | 7 | 236 | 15 |
| 165 | 19 | 29 | 15 | 289 | 20 | 334 | 17 | 253 | 9 | 200 | 18 | 282 | 14 | 318 | 7 | 245 | 15 |
| 211 | 19 | 38 | 15 | 290 | 20 | 363 | 17 | 334 | 9 | 219 | 18 | 293 | 14 | 319 | 7 | 276 | 15 |
| 220 | 19 | 47 | 15 | 292 | 20 | 1 | 16 | 335 | 9 | 257 | 18 | 300 | 14 | 340 | 7 | 307 | 15 |
| 259 | 19 | 194 | 15 | 293 | 20 | 96 | 16 | 373 | 9 | 259 | 18 | 308 | 14 | 342 | 7 | 363 | 15 |
| 269 | 19 | 219 | 15 | 295 | 20 | 98 | 16 | 15 | 9 | 285 | 18 | 333 | 14 | 343 | 7 | 15 | 14 |
| 272 | 19 | 284 | 14 | 297 | 20 | 140 | 16 | 20 | 8 | 35 | 17 | 368 | 14 | 344 | 7 | 23 | 14 |
| 273 | 19 | 339 | 14 | 302 | 20 | 142 | 16 | 39 | 8 | 41 | 17 | 19 | 13 | 359 | 7 | 107 | 14 |

**Figure A5.** SVMRFE and RFFS results of FNC data, Part 1.

| SVMRFE | | RFFS1 | | RFFS2 | | SVMRFE | | RFFS1 | | RFFS2 | | SVMRFE | | RFFS1 | | RFFS2 | |
|---|---|---|---|---|---|---|---|---|---|---|---|---|---|---|---|---|---|
| Fea | Fre | Fea | Fre | Fea | Fre | Fea | Fre | Fea | Fre | Fea | Fre | Fea | Fre | Fea | Fre | Fea | Fre |
| 30 | 13 | 367 | 7 | 108 | 14 | 247 | 11 | 282 | 5 | 26 | 11 | 216 | 9 | 92 | 2 | 178 | 9 |
| 41 | 13 | 370 | 7 | 137 | 14 | 250 | 11 | 283 | 5 | 46 | 11 | 221 | 9 | 95 | 2 | 184 | 9 |
| 56 | 13 | 375 | 7 | 161 | 14 | 260 | 11 | 287 | 5 | 51 | 11 | 236 | 9 | 96 | 2 | 216 | 9 |
| 60 | 13 | 8 | 6 | 167 | 14 | 288 | 11 | 291 | 5 | 69 | 11 | 253 | 9 | 99 | 2 | 274 | 9 |
| 132 | 13 | 23 | 6 | 181 | 14 | 290 | 11 | 296 | 5 | 72 | 11 | 291 | 9 | 103 | 2 | 319 | 9 |
| 157 | 13 | 26 | 6 | 193 | 14 | 298 | 11 | 299 | 5 | 90 | 11 | 307 | 9 | 104 | 2 | 349 | 9 |
| 188 | 13 | 34 | 6 | 209 | 14 | 302 | 11 | 307 | 5 | 124 | 11 | 316 | 9 | 108 | 2 | 370 | 9 |
| 210 | 13 | 44 | 6 | 223 | 14 | 306 | 11 | 311 | 4 | 125 | 11 | 324 | 9 | 109 | 2 | 7 | 8 |
| 268 | 13 | 58 | 6 | 329 | 14 | 375 | 11 | 316 | 4 | 133 | 11 | 349 | 9 | 110 | 2 | 55 | 8 |
| 297 | 13 | 67 | 6 | 330 | 14 | 23 | 10 | 320 | 4 | 180 | 11 | 366 | 9 | 111 | 2 | 57 | 8 |
| 304 | 13 | 73 | 6 | 371 | 14 | 25 | 10 | 322 | 4 | 234 | 11 | 12 | 8 | 116 | 2 | 76 | 8 |
| 318 | 13 | 75 | 6 | 11 | 13 | 93 | 10 | 338 | 4 | 255 | 11 | 26 | 8 | 120 | 2 | 116 | 8 |
| 325 | 13 | 83 | 6 | 95 | 13 | 139 | 10 | 347 | 4 | 294 | 11 | 34 | 8 | 121 | 2 | 134 | 8 |
| 336 | 13 | 93 | 6 | 112 | 13 | 143 | 10 | 350 | 4 | 299 | 11 | 35 | 8 | 122 | 2 | 136 | 8 |
| 50 | 12 | 97 | 6 | 191 | 13 | 175 | 10 | 355 | 4 | 311 | 11 | 73 | 8 | 126 | 2 | 192 | 8 |
| 54 | 12 | 100 | 6 | 248 | 13 | 180 | 10 | 357 | 4 | 340 | 11 | 99 | 8 | 127 | 2 | 197 | 8 |
| 55 | 12 | 107 | 6 | 253 | 13 | 196 | 10 | 363 | 4 | 344 | 11 | 117 | 8 | 133 | 2 | 250 | 8 |
| 58 | 12 | 113 | 5 | 260 | 13 | 202 | 10 | 365 | 4 | 345 | 11 | 129 | 8 | 135 | 2 | 271 | 8 |
| 71 | 12 | 115 | 5 | 277 | 13 | 212 | 10 | 372 | 4 | 365 | 11 | 145 | 8 | 136 | 2 | 287 | 8 |
| 127 | 12 | 123 | 5 | 282 | 13 | 225 | 10 | 376 | 4 | 373 | 11 | 191 | 8 | 138 | 2 | 308 | 8 |
| 141 | 12 | 130 | 5 | 291 | 13 | 227 | 10 | 2 | 4 | 375 | 11 | 197 | 8 | 147 | 2 | 316 | 8 |
| 148 | 12 | 131 | 5 | 313 | 13 | 230 | 10 | 3 | 4 | 34 | 10 | 209 | 8 | 155 | 2 | 361 | 8 |
| 218 | 12 | 137 | 5 | 315 | 13 | 263 | 10 | 6 | 4 | 44 | 10 | 214 | 8 | 158 | 2 | 367 | 8 |
| 223 | 12 | 141 | 5 | 22 | 12 | 280 | 10 | 11 | 4 | 81 | 10 | 229 | 8 | 160 | 2 | 8 | 7 |
| 357 | 12 | 145 | 5 | 39 | 12 | 309 | 10 | 12 | 3 | 82 | 10 | 255 | 8 | 161 | 2 | 58 | 7 |
| 14 | 11 | 151 | 5 | 66 | 12 | 315 | 10 | 17 | 3 | 130 | 10 | 266 | 8 | 163 | 2 | 103 | 7 |
| 17 | 11 | 152 | 5 | 109 | 12 | 346 | 10 | 18 | 3 | 222 | 10 | 294 | 8 | 164 | 2 | 104 | 7 |
| 51 | 11 | 159 | 5 | 114 | 12 | 354 | 10 | 21 | 3 | 238 | 10 | 296 | 8 | 167 | 2 | 111 | 7 |
| 63 | 11 | 162 | 5 | 127 | 12 | 4 | 9 | 31 | 3 | 239 | 10 | 344 | 8 | 177 | 2 | 145 | 7 |
| 64 | 11 | 166 | 5 | 131 | 12 | 6 | 9 | 32 | 3 | 242 | 10 | 351 | 8 | 179 | 2 | 146 | 7 |
| 68 | 11 | 168 | 5 | 144 | 12 | 9 | 9 | 35 | 3 | 249 | 10 | 355 | 8 | 184 | 2 | 155 | 7 |
| 112 | 11 | 176 | 5 | 186 | 12 | 28 | 9 | 36 | 3 | 273 | 10 | 378 | 8 | 186 | 2 | 158 | 7 |
| 167 | 11 | 180 | 5 | 198 | 12 | 46 | 9 | 43 | 3 | 305 | 10 | 49 | 7 | 187 | 2 | 164 | 7 |
| 169 | 11 | 199 | 5 | 212 | 12 | 47 | 9 | 46 | 3 | 306 | 10 | 65 | 7 | 195 | 2 | 176 | 7 |
| 173 | 11 | 215 | 5 | 225 | 12 | 76 | 9 | 50 | 2 | 372 | 10 | 69 | 7 | 202 | 1 | 214 | 7 |
| 205 | 11 | 216 | 5 | 246 | 12 | 95 | 9 | 53 | 2 | 36 | 9 | 77 | 7 | 204 | 1 | 229 | 7 |
| 207 | 11 | 238 | 5 | 263 | 12 | 101 | 9 | 56 | 2 | 50 | 9 | 85 | 7 | 205 | 1 | 262 | 7 |
| 208 | 11 | 241 | 5 | 288 | 12 | 119 | 9 | 72 | 2 | 86 | 9 | 91 | 7 | 207 | 1 | 268 | 7 |
| 226 | 11 | 256 | 5 | 309 | 12 | 170 | 9 | 76 | 2 | 139 | 9 | 116 | 7 | 212 | 1 | 269 | 7 |
| 231 | 11 | 260 | 5 | 358 | 12 | 184 | 9 | 82 | 2 | 141 | 9 | 126 | 7 | 217 | 1 | 270 | 7 |
| 237 | 11 | 262 | 5 | 374 | 12 | 203 | 9 | 84 | 2 | 159 | 9 | 146 | 7 | 218 | 1 | 272 | 7 |
| 239 | 11 | 263 | 5 | 6 | 11 | 204 | 9 | 87 | 2 | 175 | 9 | 176 | 7 | 230 | 1 | 300 | 7 |

**Figure A6.** SVMRFE and RFFS results of FNC data, Part 2.

| SVMRFE | | RFFS1 | | RFFS2 | | SVMRFE | | RFFS1 | | RFFS2 | | SVMRFE | | RFFS1 | | RFFS2 | |
|---|---|---|---|---|---|---|---|---|---|---|---|---|---|---|---|---|---|
| Fea | Fre | Fea | Fre | Fea | Fre | Fea | Fre | Fea | Fre | Fea | Fre | Fea | Fre | Fea | Fre | Fea | Fre |
| 186 | 7 | 231 | 1 | 320 | 7 | 108 | 5 | 7 | 0 | 2 | 4 | 3 | 3 | 175 | 0 | 207 | 2 |
| 195 | 7 | 235 | 1 | 327 | 7 | 111 | 5 | 10 | 0 | 52 | 4 | 31 | 3 | 178 | 0 | 224 | 2 |
| 217 | 7 | 237 | 1 | 343 | 7 | 113 | 5 | 14 | 0 | 70 | 4 | 36 | 3 | 188 | 0 | 240 | 2 |
| 241 | 7 | 239 | 1 | 351 | 7 | 121 | 5 | 16 | 0 | 73 | 4 | 52 | 3 | 191 | 0 | 241 | 2 |
| 248 | 7 | 240 | 1 | 14 | 6 | 131 | 5 | 19 | 0 | 80 | 4 | 80 | 3 | 192 | 0 | 258 | 2 |
| 258 | 7 | 249 | 1 | 17 | 6 | 133 | 5 | 22 | 0 | 94 | 4 | 82 | 3 | 196 | 0 | 283 | 2 |
| 267 | 7 | 250 | 1 | 24 | 6 | 137 | 5 | 24 | 0 | 140 | 4 | 84 | 3 | 197 | 0 | 310 | 2 |
| 274 | 7 | 254 | 1 | 28 | 6 | 149 | 5 | 25 | 0 | 147 | 4 | 92 | 3 | 201 | 0 | 331 | 2 |
| 283 | 7 | 258 | 1 | 93 | 6 | 150 | 5 | 27 | 0 | 154 | 4 | 105 | 3 | 203 | 0 | 338 | 2 |
| 284 | 7 | 266 | 1 | 100 | 6 | 156 | 5 | 28 | 0 | 166 | 4 | 120 | 3 | 206 | 0 | 348 | 2 |
| 286 | 7 | 267 | 1 | 115 | 6 | 159 | 5 | 52 | 0 | 177 | 4 | 123 | 3 | 214 | 0 | 364 | 2 |
| 319 | 7 | 269 | 1 | 152 | 6 | 160 | 5 | 55 | 0 | 187 | 4 | 128 | 3 | 223 | 0 | 369 | 2 |
| 350 | 7 | 270 | 1 | 188 | 6 | 163 | 5 | 57 | 0 | 203 | 4 | 134 | 3 | 224 | 0 | 378 | 2 |
| 376 | 7 | 281 | 1 | 201 | 6 | 166 | 5 | 59 | 0 | 247 | 4 | 136 | 3 | 225 | 0 | 18 | 1 |
| 377 | 7 | 285 | 1 | 227 | 6 | 192 | 5 | 60 | 0 | 280 | 4 | 164 | 3 | 227 | 0 | 56 | 1 |
| 7 | 6 | 286 | 1 | 261 | 6 | 246 | 5 | 65 | 0 | 354 | 4 | 245 | 3 | 228 | 0 | 79 | 1 |
| 15 | 6 | 288 | 1 | 267 | 6 | 281 | 5 | 66 | 0 | 377 | 4 | 249 | 3 | 229 | 0 | 87 | 1 |
| 27 | 6 | 298 | 1 | 324 | 6 | 303 | 5 | 70 | 0 | 19 | 3 | 257 | 3 | 236 | 0 | 97 | 1 |
| 39 | 6 | 301 | 1 | 346 | 6 | 317 | 5 | 74 | 0 | 27 | 3 | 264 | 3 | 247 | 0 | 99 | 1 |
| 66 | 6 | 303 | 1 | 360 | 6 | 348 | 5 | 77 | 0 | 32 | 3 | 287 | 3 | 252 | 0 | 120 | 1 |
| 87 | 6 | 305 | 1 | 362 | 6 | 369 | 5 | 79 | 0 | 84 | 3 | 327 | 3 | 261 | 0 | 121 | 1 |
| 104 | 6 | 313 | 1 | 12 | 5 | 370 | 5 | 80 | 0 | 92 | 3 | 360 | 3 | 271 | 0 | 143 | 1 |
| 107 | 6 | 314 | 1 | 31 | 5 | 374 | 5 | 86 | 0 | 110 | 3 | 362 | 3 | 272 | 0 | 148 | 1 |
| 114 | 6 | 315 | 1 | 60 | 5 | 10 | 4 | 91 | 0 | 122 | 3 | 365 | 3 | 273 | 0 | 160 | 1 |
| 135 | 6 | 317 | 1 | 65 | 5 | 21 | 4 | 94 | 0 | 126 | 3 | 367 | 3 | 280 | 0 | 196 | 1 |
| 151 | 6 | 325 | 1 | 74 | 5 | 22 | 4 | 101 | 0 | 129 | 3 | 11 | 2 | 294 | 0 | 231 | 1 |
| 154 | 6 | 329 | 1 | 98 | 5 | 32 | 4 | 105 | 0 | 168 | 3 | 44 | 2 | 300 | 0 | 266 | 1 |
| 161 | 6 | 330 | 1 | 101 | 5 | 81 | 4 | 112 | 0 | 206 | 3 | 53 | 2 | 308 | 0 | 326 | 1 |
| 172 | 6 | 341 | 1 | 117 | 5 | 88 | 4 | 114 | 0 | 252 | 3 | 67 | 2 | 310 | 0 | 341 | 1 |
| 178 | 6 | 346 | 1 | 119 | 5 | 122 | 4 | 117 | 0 | 281 | 3 | 72 | 2 | 324 | 0 | 357 | 1 |
| 187 | 6 | 348 | 1 | 174 | 5 | 130 | 4 | 124 | 0 | 314 | 3 | 97 | 2 | 326 | 0 | 1 | 0 |
| 206 | 6 | 349 | 1 | 195 | 5 | 155 | 4 | 129 | 0 | 332 | 3 | 103 | 2 | 327 | 0 | 3 | 0 |
| 240 | 6 | 351 | 1 | 204 | 5 | 168 | 4 | 132 | 0 | 10 | 2 | 153 | 2 | 331 | 0 | 77 | 0 |
| 261 | 6 | 354 | 1 | 218 | 5 | 174 | 4 | 134 | 0 | 21 | 2 | 224 | 2 | 332 | 0 | 123 | 0 |
| 320 | 6 | 358 | 1 | 237 | 5 | 222 | 4 | 139 | 0 | 25 | 2 | 228 | 2 | 336 | 0 | 132 | 0 |
| 323 | 6 | 360 | 1 | 254 | 5 | 271 | 4 | 140 | 0 | 53 | 2 | 252 | 2 | 345 | 0 | 153 | 0 |
| 347 | 6 | 361 | 1 | 303 | 5 | 310 | 4 | 143 | 0 | 59 | 2 | 326 | 2 | 352 | 0 | 202 | 0 |
| 359 | 6 | 362 | 1 | 317 | 5 | 314 | 4 | 146 | 0 | 67 | 2 | 358 | 2 | 356 | 0 | 217 | 0 |
| 16 | 5 | 364 | 1 | 322 | 5 | 321 | 4 | 149 | 0 | 91 | 2 | 86 | 1 | 366 | 0 | 230 | 0 |
| 74 | 5 | 369 | 1 | 352 | 5 | 335 | 4 | 153 | 0 | 149 | 2 | 109 | 1 | 371 | 0 | 286 | 0 |
| 90 | 5 | 378 | 1 | 355 | 5 | 343 | 4 | 154 | 0 | 162 | 2 | 115 | 1 | 374 | 0 | 336 | 0 |
| 94 | 5 | 1 | 0 | 359 | 5 | 364 | 4 | 174 | 0 | 199 | 2 | 330 | 1 | 377 | 0 | 366 | 0 |

**Figure A7.** SVMRFE and RFFS results of FNC data, Part 3.

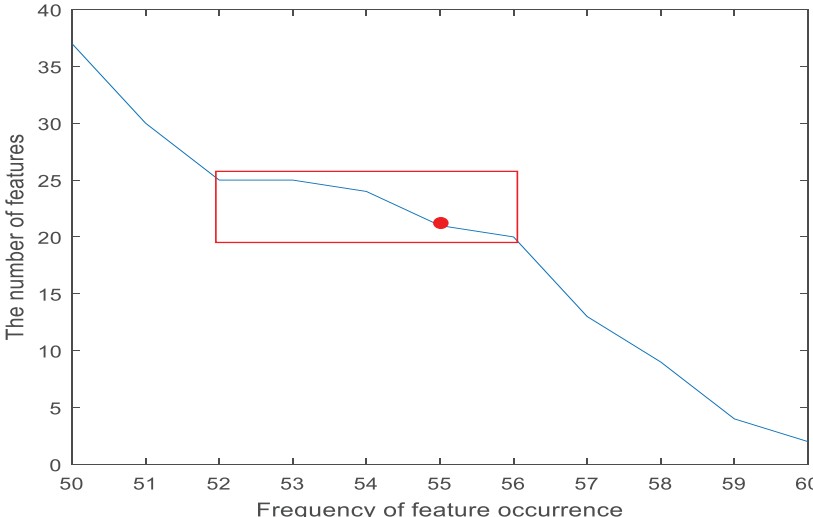

**Figure A8.** The characteristic frequency distribution with a frequency greater than or equal to 50. The *x* axis corresponds to the frequency of occurrence, and the *y* axis is the number of features. We can find that when the frequency is in the red range, i.e., greater than or equal to 52 and less than or equal to 56, the number of features is quite stable. Compared with other ranges, in the red range, there exists a balance between the number of features and the frequency of occurrence, which facilitates the abnormal analysis of brain function connections and structures corresponding to diseases. Therefore, we selected features with a frequency greater than or equal to 55.

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
