# Peer review of "Identifying Brain Abnormalities with Schizophrenia Based on a Hybrid Feature Selection Technology"

_applsci, doi:10.3390/app9102148_

Round 1

Reviewer 1 Report

Please find my comments in the attachment.

Author Response

Point 1: The description: Wondering if any correction for multiple comparisons was considered while employing statistics, two-test analysis, as shown in Figure 6 and reported in Table 1.

Response 1: Sincerely here, we quite appreciate your affirmation of our paper. In the original manuscript, we only used the p-values of hypothesis test for statistics analysis, and based on your insightful suggestions, we have calculated the FDR correction of p-values to support the significance of the features obtained by the proposed hybrid feature selection method. The results show that most of the finally selected features have quite notable FDR p-values, especially for all of the SBMs data. Although some FDR p-values of the FNC data are not quite small, i.e., for the 40th, the 165th and the 185th features, the FDR p-values of them are over 0.05, while, when sort the FDR p-values of all the original features, they are still at the front of the queue. Additionally, since these features are selected by all the six kinds of machine learning methods and statistical methods, we have reason to believe that these features should be considered as the possible potential anomalies of functional connections existed in the SZ’s brain. The further analysis on the corresponding functional connectivity of brains show that these selected connections are significant related to SZ. For example, the 40th feature is the connection between the rolandic operculum (ROL) and the superior occipital gyrus (SOG). Current researches have shown that the reduction of connectivity of ROL is related to language and hallucination. As we know, the disorder of language and hallucination are the main clinical manifestations of SZ. The 165th feature is the connection between the insula (INS) and the amygdala (AMYG). Studies have found that the decreasing connections with INS may cause the disrupted functional integration of brains. The 185th feature is the connection between the caudate nucleus (CAU) and SOG, and related researches have revealed that the abnormal connectivity related to CAU can affect the cognitive function of human beings, resulting in decreased memory ability, and it may be the cause of cognitive deficits in SZ. More explanations of all the selected abnormal functional connections related with SZ can be found in the Discussion part of the manuscript. Thank you.

Point 2: Figure 7. Details in the caption are incomplete. Please add details on how the cortical maps were related to the features. Figure 8. Details in the figure caption are incomplete. Please add units and legend to the circular connectivity graph. Please add full names of ML and SM to the figure table.

Response 2: Thanks for your good comments. According to your suggestions, we have added a reference of the relationships between the cortical maps and the features, and the details can be found in the caption of Figure 7. We also added the corresponding explanations of the circular connectivity graph, the full names of ML and SM, and some details of Figure 8 in the caption of it.

Point 3: Line 348. Please revise the sentence as, "Further to support the validity of connectivity findings..." [There is no ground truth, so full verification is not feasible.]

Response 3: We really appreciate your kind suggestions. To overcome the inaccuracy of the sentence “To further verify the validation of the abnormal functional connections selected by the above…” in the original manuscript, in the revised paper, we have revised the sentence in line 348 (now 349) as “Further to support the validity of connectivity findings by the above...”.

In all, we want to show our great gratitude to your suggestions and comments, which are very valuable in improving the quality of our manuscript. Thank you again for your warm help!

Reviewer 2 Report

General

The paper proposes a method to explore brain abnormalities by SZ. MRI data are utilized, and the method proposed demonstrates abnormal brain regions and abnormal disruptions of the brain network in SZ. The study is scientifically sound, and the paper is in general well-written, therefore it can be considered for publication in Appl. Sc., with several improvements, as suggested bellow.

Specific

1) English is in general fine, but some edits and flaws do exist, for example: the phrase in lines 30-32 has no sense, it has to be reformulated; do not start a sentence with a number, as in line 34; once an acronym introduced, please continue to use it, e.g. SZ in line 42, etc.; ‘can be found out’ in line 61; ‘are be combined’ in line 81, etc. Please check the entire manuscript for errors and correct them.

2)  Please explain for the readers (especially in the Abstract) statements such as “the sample size is usually smaller than the dimension”. Abstract, Introduction, and Conclusions must be readable by non-specialists.

Author Response

Point 1: English is in general fine, but some edits and flaws do exist, for example: the phrase in lines 30-32 has no sense, it has to be reformulated; do not start a sentence with a number, as in line 34; once an acronym introduced, please continue to use it, e.g. SZ in line 42, etc.; ‘can be found out’ in line 61; ‘are be combined’ in line 81, etc. Please check the entire manuscript for errors and correct them.

Response 1: It is very kind of you to point out the mistakes. We have tried our best to correct the edit as well as syntax errors in the revised paper and hope that these errors no longer exist. For example, the sentence in lines 30-32 has been reformulated to “Du et al. applied a novel group information guided method to estimate inherent dynamic functional brain networks and found that the abnormalities of SZ were mainly distributed in the cerebellum, frontal cortex, thalamus and temporal cortex [4]”. In line 34, the sentence has been revised as “In [6], Rosenberg et al. demonstrated that…”. The acronyms, such as SZ for schizophrenia, have been checked carefully and in the entire manuscript, they are continuously used after the first introduction of them. The sentence in line 61 (now in line 62) has been revised as “can be found out”. In line 81 (now in line 82), the sentence has been modified as “...can be combined...”. Some other typos and writing mistakes have also been corrected.   

Point 2:  Please explain for the readers (especially in the Abstract) statements such as “the sample size is usually smaller than the dimension”. Abstract, Introduction, and Conclusions must be readable by non-specialists.

Response 2: We really appreciate your insightful comments. You are quite right, in the original manuscript, there existed some unclear parts in the Abstract, Introduction and Conclusions. So, we have made some improvements on the original manuscript. For example, in the Abstract, in line 2, the sentence has been revised as “are usually of small sample size but large number of features”. In lines 44-45 of the Introduction, we added “However, for MRI data of SZ, they are usually of small sample size but large number of features, i.e., n<<p, where n is the sample size and p is the number of features” to further explain that for MRI data of SZ, the sample size is usually smaller than the dimension of features. In line 436 of the Conclusions, we added “All of the results suggest that the brain regions and connectivity in SZ are destroyed compared with HCs” to make the sentence to be more readable. In addition, in order to make the expression more precise, in line 438, we changed the sentence “All findings ensure the validation of the proposed…" to be "All findings support the validation of the proposed…”.

In all, we want to show our great gratitude to your suggestions and comments, which are very valuable in improving the quality of our manuscript. Thank you again for your warm help!
